# Hydrological Forecasting under Climate Variability Using Modeling and Earth Observations in the Naryn River Basin, Kyrgyzstan

**Merim Pamirbek kyzy** [1,2,3,4,5], **Xi Chen** [1,2,3,*], **Tie Liu** [1,2,3], **Eldiiar Duulatov** [4,6], **Akmal Gafurov** [7], **Elvira Omorova** [8] and **Abror Gafurov** [9]

1   State Key Laboratory of Desert and Oasis Ecology, Xinjiang Institute of Ecology and Geography, Chinese Academy of Sciences, Urumqi 830011, China
2   Research Center for Ecology and Environment of Central Asia, Chinese Academy of Sciences, Urumqi 830011, China
3   University of Chinese Academy of Sciences, Beijing 100049, China
4   Institute of Geology, National Academy of Sciences of the Kyrgyz Republic, Bishkek 720040, Kyrgyzstan
5   Institute of Water Problems and Hydropower, National Academy of Science of the Kyrgyz Republic, Bish-kek 720033, Kyrgyzstan
6   Research Center for Ecology and Environment of Central Asia (Bishkek), Bishkek 720040, Kyrgyzstan
7   Hydrometeorological Service of the Republic of Uzbekistan, Tashkent 100052, Uzbekistan
8   Hydrometeorological Service under the Ministry of Emergency Situations of the Kyrgyz Republic, Bishkek 720017, Kyrgyzstan
9   Section Hydrology, GFZ German Research Centre for Geosciences (GFZ Potsdam), 14473 Potsdam, Germany
*   Correspondence: chenxi@ms.xjb.ac.cn

**Abstract:** The availability of water resources in Central Asia depends greatly on snow accumulation in the mountains of Tien-Shan and Pamir. It is important to precisely forecast water availability as it is shared by several countries and has a transboundary context. The impact of climate change in this region requires improving the quality of hydrological forecasts in the Naryn river basin. This is especially true for the growing season due to the unpredictable climate behavior. A real-time monitoring and forecasting system based on hydrological watershed models is widely used for forecast monitoring. The study's main objective is to simulate hydrological forecasts for three different hydrological stations (Uch-Terek, Naryn, and Big-Naryn) located in the Naryn river basin, the main water formation area of the Syrdarya River. We used the MODSNOW model to generate statistical forecast models. The model simulates the hydrological cycle using standard meteorological data, discharge data, and remote sensing data based on the MODIS snow cover area. As for the forecast at the monthly scale, the model considers the snow cover conditions at separate elevation zones. The operation of a watershed model includes the effects of climate change on river dynamics, especially snowfall and its melting processes in different altitude zones of the Naryn river basin. The linear regression models were produced for monthly and yearly hydrological forecasts. The linear regression shows $R^2$ values of 0.81, 0.75, and 0.77 (Uch-Terek, Naryn, and Big-Naryn, respectively). The correlation between discharge and snow cover at various elevation zones was used to examine the relationship between snow cover and the elevation of the study. The best correlation was in May, June, and July for the elevation ranging from 1000–1500 m in station Uch-Terek, and 1500–3500 m in stations Naryn and Big-Naryn. The best correlation was in June: 0.87; 0.76; 0.84, and May for the elevation ranging from 1000–3500 m in station Uch-Terek, and 2000–3000 m in stations Naryn and Big-Naryn. Hydrological forecast modeling in this study aims to provide helpful information to improve our under-standing that the snow cover is the central aspect of water accumulation.

**Keywords:** MODSNOW-Tool; hydrological forecast; snow cover area; MODIS; Naryn river basin

## 1. Introduction

Seasonal snow cover is one of the most critical components affecting the surface energy balance and hydrological balance [1,2]. Seasonal snow provides the predominant contribution to the flow of large rivers in high-altitude areas [3]. Snowmelt mainly dominates spring and early summer season runoff [4] and plays a vital role in the water supply of various sectors [5,6]. In addition, climate change affects seasonality, snowmelt onset, and water availability during the summer season [7,8]. It is crucial to understand such changes to manage water resources and prepare for extreme events effectively [9], such as floods and droughts [10]. The vast spatial extent and topographic complexity of snow-capped mountains make field monitoring difficult [11]. Therefore, remote sensing is the most suitable source of observation for snow conditions in remote areas. Remote sensing snow products play an essential role in hydrological and other snow-related studies [12]. Observations from spatially distributed snow cover space significantly improve hydrological models of runoff forecasting. Modeling the accumulation of water using the snow cover and the spread of snow cover over high-altitude zones is already accepted in many countries and improves the validity of forecasts. Remote sensing-based snow cover data are mainly used for: water balance [13]; runoff during different seasons [11]; runoff during high flow conditions; runoff during low flow conditions [14]; spring flood (volume, peak); peak runoff during autumn [5,15], and spring river discharge [16]. Traditionally, data from snow surveys and snow observations at weather stations in foothills and mountainous areas are used to predict the flow of Mountain Rivers in Central Asia [17]. However, such surveys and observations are limited at high altitudes due to the inaccessibility of remote areas. Satellite images provide us with visual and quantitative information and allow us to monitor the spatially distributed snow cover in remote areas. Considering the required knowledge to process satellite images and the unavailability of capacities at hydro-meteorological services, the automation and digitalization of these procedures are of great relevance to monitoring snow cover in mountainous regions and applying for hydrological forecasts. Observations from the space is very convenient method for hydrological forecasting in the high mountain areas in Central Asia [18].

However, optical sensors can detect only snow cover in areas with clear sky conditions. Therefore, it is vital to accurately assess the presence of snow under clouds and reduce or eliminate cloud cover from MODIS snow cover images. For this purpose, the MODSNOW software package was developed and adapted for Central Asia [19]. Traditionally, data from snow surveys and snow observations at weather stations in foothills and mountainous areas are used to predict the flow of Mountain Rivers in Central Asia [7]. Unfortunately, the method is old and difficult to reach in the wintertime in high mountains areas, and due to climate change, new approaches to hydrological forecasting are needed. Moreover, more weather stations in the country are not working now and need reconstruction. The MODSNOW results provide visual and quantitative information on snow coverage and allow us to monitor the spatial distribution of snow cover in remote areas. Daily snow cover data from MODSNOW can provide a thorough assessment of water accumulation in the snow cover, taking into account the altitude, local temperature, and the snow melting process in each sub-basin.

Previous studies with spatiotemporal data demonstrate the significant correlation between topographic shading and mass balance in the glaciers of Central Asia [20]. Another study is being carried out to show the spatial and temporal variations in snow cover for the period 2000–2015 [21]. However, mountainous regions revealed that altitude was a predominant factor influencing snowpack melting [22]. Snow cover monitoring for highland pastures in Central Asia and long-term snow depth dynamics patterns may be useful for water resource management and river discharge planning. The elevation zone is an important factor for the snowmelt process and assessment. The snow melting process can improve the quality of the hydrological forecast in the Naryn river basin, considering all influencing factors such as local temperature, precipitation, and geographical conditions. The study aims to answer the question of making a hydrological forecast for the monthly

river flow of the Naryn river basin and the growing season using snow cover based on remote sensing data and hydrometeorological data. Research objectives: to determine the dynamics of snowmelt in altitudinal zones, analyze spatiotemporal changes in snow cover, explore potential relationships between topographic factors and climatic parameters, as well as the relationship between snow cover area and meteorological factors (precipitation, temperature) to make hydrological forecasts.

## 2. Materials and Methods

### 2.1. Study Area

The Naryn river basin (NRB) is located in the Central Tien Shan mountain range of Kyrgyzstan, the headwaters of the Syrdarya River. The length of the Naryn River in Kyrgyzstan is about 600 km, and the basin area is about 60.000 km², with elevations ranging from 868 m above sea level (m.a.s.l.) to 5135 m.a.s.l. (Figure 1). In the NRB, temperatures typically range from −12 °C to 25 °C throughout the year and are rarely below −17 °C or above 30 °C. Winters are very cold and snowy. The snow cover lasts for 180–200 days, and its height reaches 1 m in some places. The annual amount of precipitation is 270–280 mm, and 75% of it falls during the warm period. The maximum precipitation occurs in May–June, and the minimum occurs in January–December. The average long-term annual water discharge in the big Naryn gauge is 47.1 m³/s; 92.9 m³/s at the Naryn gauge; and 326 m³/s at Uch-Terek gauge [23].

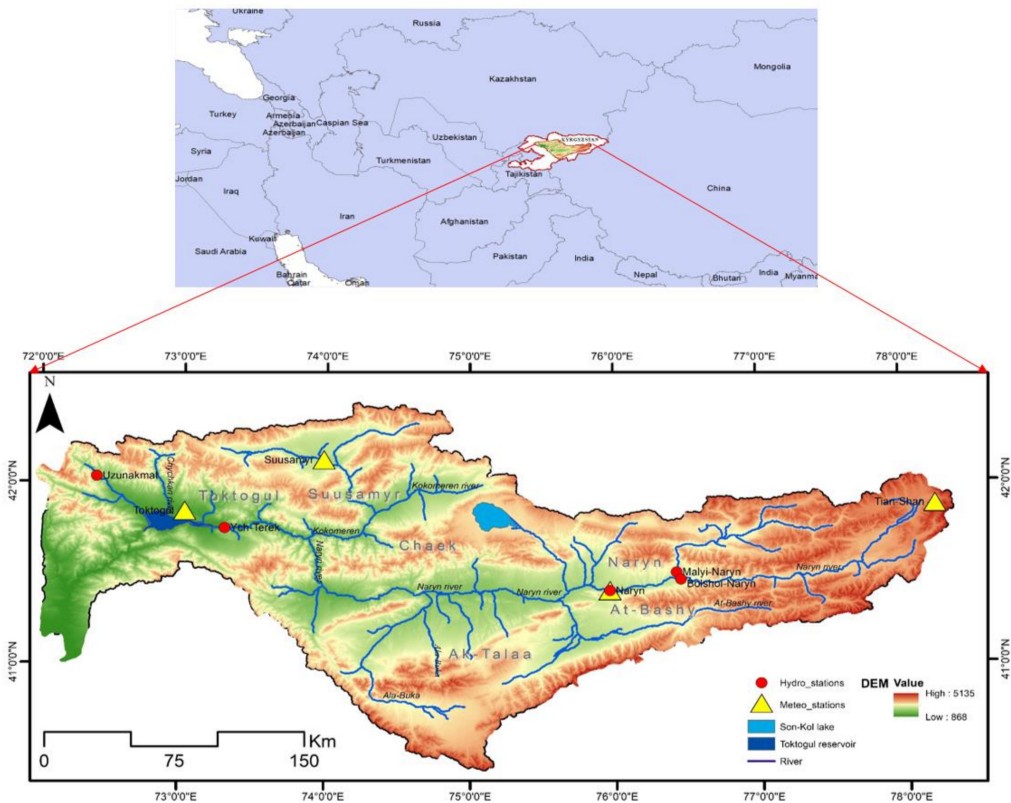

**Figure 1.** The map of the Naryn river basin.

The upper reaches of the river freeze in the winter, and peak flow occurs from May to June. The basin has four meteorological stations and six hydro stations. In this study, the entire Naryn river basin was partitioned into three regions according to the distributional and topographic features of the source streams: Upstream, where the Tian-Shan meteorological station and Big-Naryn hydrological stations are located. In the midstream are the hydrometeorological stations of Naryn, the Toktogul meteorological station, and the Uch-Terek hydrological station is on the lower stream.

Figure 2 shows the water evaluation of the Naryn river basin, which is necessary to determine the peak flow season, as well as to determine the forecast period. The graphs show the beginning and end of the snow melting period and the river discharge dynamics accordingly. As shown in Figure 2, the snow cover amount is important during winter due to the low temperature and consequent snow accumulation, leading to a total absence of the discharge. In contrast, the temperature increases during summer, resulting in snow melting. During the spring and summer months, the increase in discharge (Q) was triggered by snow melting. However, with the negligible flow during winter, it is useless to forecast water availability.

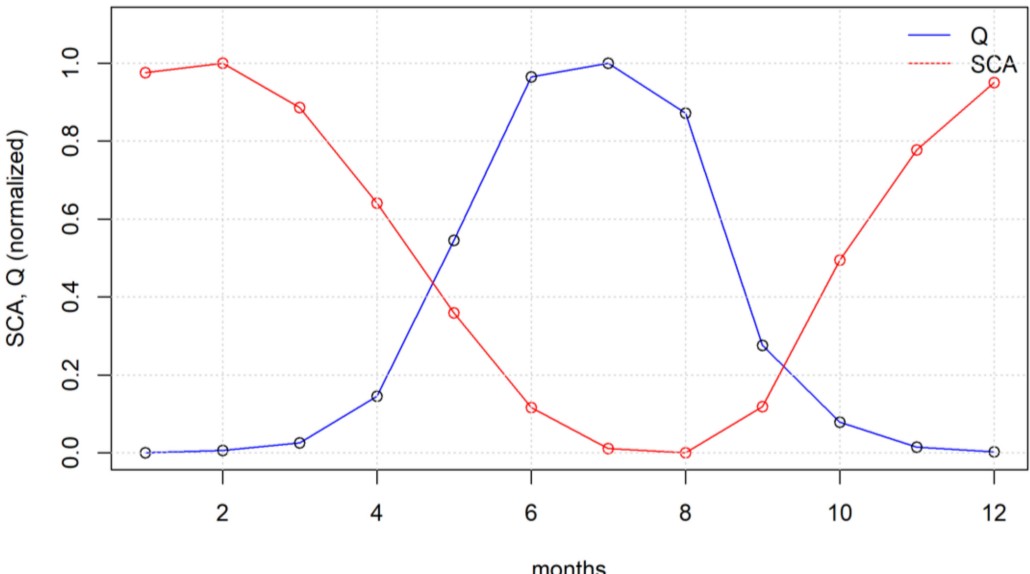

**Figure 2.** Discharge and Snow cover evaluation of Naryn river basin. The data is averaged from 3 hydrological stations (Big-Naryn, Naryn, Uch-Terek 2000–2021).

*2.2. Data Used*

In the Naryn river basin, snow cover plays a significant role because from October to early April, the snow cover is accumulated and starts to melt from early spring and directly forms the river runoff. Furthermore, according to the hydrological calendar, April to October is the growing season [24]. Therefore, snow cover is considered the main predictor of the hydrological forecast model in this study. The MODSNOW processes snow cover data from MODIS [25], including cloud removal, and creates a daily cloudless snow cover time series from February 2000 to October 2021 for any predefined river basin [26]. After removing the cloud coverage, the MODSNOW also generates snow cover data for each altitude zone in the river basin at 500 m or 1000 m intervals. Then, the state of snow cover in different altitude zones and periods is used as a predictor for the monthly forecast of water availability. Using data on snow cover at different elevation zones, the MODSNOW then generates statistical forecast models for each month using the historical snow cover as the main predictor and the antecedent river discharge as a second predictor. In the Naryn river basin, snow melting contributes significantly (most contributing component) to the total runoff during the summer months, when precipitation is negligible. In particular, in May, June, and July, the contribution of snowmelt to the total runoff is very high. Thus, snow-related information such as time series of snow cover based on satellite data can be significantly helpful for monthly assessments of water availability [27]. The hydrological forecast for one month needs previous monthly snow cover.

The MODIS snow cover data for the period 2000–2021 was processed using the MODSNOW, and cloud-free snow cover time series were prepared to develop statistical models. Besides the MODIS snow cover data, the SRTM Digital Elevation Model (DEM) was

used in this study obtained from the Earth-Explorer database. To have the exact resolution as the MODIS data, the 90 m SRTM DEM was aggregated using ArcMap software to divide into nine elevation zones (every 500 m). The hydro-meteorological data from the meteorological and hydrological stations for the period 2000–2021. Monthly precipitation, mean monthly temperature, and monthly discharge data of the Naryn river basin were obtained from the Hydro-meteorological Service of Kyrgyzstan archive.

### 2.3. Methodology

Statistical forecast models were developed using daily snow cover data obtained from MODIS and processed through the MODSNOW-Tool, monthly river discharge data (Q), monthly average air temperature data and monthly total precipitation data for the antecedent months. We used the snow cover area (SCA) as the first predictor assuming that snow cover data to be a good predictor in snowmelt-dominated river basins. A multiple linear regression approach was used for the forecast models, as shown in Equation (1).

$$Q = a * SCA + b * Qp + c * Prec + d * Temp \tag{1}$$

where: $SCA$ is the Snow Cover Area processed using the MODSNOW-Tool (% of snow coverage relative to entire river basin area), $Q$ is the forecasted discharge ($m^3$), $a$, $b$, $c$, $d$ are the coefficients of a regression model ($m^3$), $Qp$ is the antecedent discharge ($m^3$), $Prec$ is the monthly total precipitation (mm) before forecast period, and $Temp$ is the average monthly temperature (°C) before forecast period.

A forecast model is a function that is integrated into the MODSNOW. The correlation coefficient was used to select optimal hydrological forecast models for each river basin. For hydrological forecasts in the growing season, the mean monthly snow cover area covering the entire river basin, mean monthly discharge, total monthly precipitation, and average air temperature in the months preceding the forecast was used as the predictors. For example, for the forecast model during the growth season April–September, the mean snow cover area, mean precipitation, mean temperature, and mean discharge in March were used as predictors. In the case of a monthly forecast (e.g., June), the mean snow cover area of a particular elevation in the previous month (e.g., May) was used as the main predictor. Additionally, for the monthly scale, additional predictors such as the antecedent discharge ($Qp$), total precipitation ($Prec$), and mean temperature ($Temp$) for the previous month were used as predictors. The antecedent discharge can represent the groundwater storage in the regression models and thus can also be called an inertial movement of water storage from groundwater towards the river. In other words, the antecedent discharge can represent the groundwater contribution in forecast models. However, the antecedent discharge was considered as an additional predictor if using only the snow cover area did not lead to satisfactory forecast models.

The correlation coefficient and coefficient of determination (R, $R^2$) are good performance measures to signify the consistency between observed and simulated data. This is done by following the best fit line, which is used to assess the predictive efficiency of the hydrological model [28] and quantitatively describe the accuracy between model outputs (forecasts) and observed data during the study period (2000–2021). The most common metric used to predict the model's accuracy for the selected loss and transformation methods is the accuracy between observed and simulated values, defined between 0–1. The maximum possible value of the objective function is 1, which indicates a complete correspondence between the simulated and observed flow [29]. Since they are susceptible to high flow rates, calculative costs are more sensitive at low costs [30]. This analysis is defined by criteria using stepwise regression analysis to link the observed and simulated data, following Equation (2).

At the bottom of the formula for the correlation coefficient (R), if this value is squared, ($R^2$) becomes the coefficient of determination.

In the equation, *x*-independent parameter (for example, snow), *y*-dependent parameter (for example, river discharge), *n* is the number of observations, or years, where data is available and used in the formula $\sum$ = Summation using Equation (2).

$$r = \frac{n(\sum xy) - (\sum x)(\sum x)(\sum y)}{\sqrt{\left[n\sum x^2 - (\sum x)^2\right]\left[n\sum y^2 - (\sum y)^2\right]}} \tag{2}$$

For the development of a new forecast models using the multiple linear regression approach, the historical data on snow cover area, antecedent discharge, total monthly precipitation, and average monthly temperature can be prepared and, using the MOD-SNOW forecast functions, the new multiple linear regression models can be developed automatically.

We also performed a validation of forecast models by skipping each of historical years and forecasting water availability for those particular years and compared again observed values. This procedure is also done automatically by fully integrated forecast function into the MODSNOW.

## 3. Results

### 3.1. Simulation Results of MODSNOW

Figure 3 shows the snow cover area evolution for each hydrological year processed using the MODSNOW-Tool. It is also demonstrated that there is a high variation of snow cover area at the end of March and April, which corresponds to water availability in those years and can thus be used as a valuable predictor for hydrological forecasts modeling as a snowmelt-dominated river discharge like in Naryn river basin. The snow cover area in the lowest elevation accumulates at the latest and starts to melt at the earliest due to higher temperatures in the respective elevation. Consequently, the highest elevation zone holds more snow during the entire year than lower elevation zones. As a result, the melting snow cover will start later than at lower elevations, which regulates the formation of an uninterrupted annual river flow.

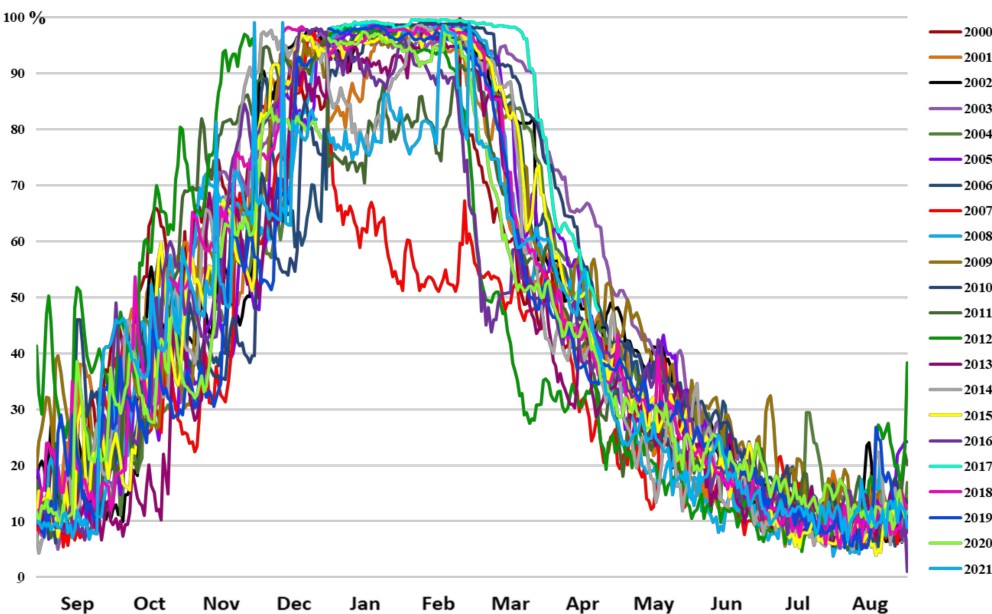

**Figure 3.** The evolution of daily snow covers the hydrological year 2000 to 2021.

The snowmelt season in early summer is vulnerable to hydrological extremes, particularly to floods, mudflows, and landslides that need such kind of monitoring in the Naryn

river basin. Having subdivided the watershed into elevation zones and snow cover area for the hydrological year, as illustrated in Figure 4, allows developing predictors for monthly scale hydrological forecasting and also shows the elevation-dependent evolution of snow cover area obtained from MODSNOW software and reveals the comparative analysis of snow cover evolution of the Naryn river basin.

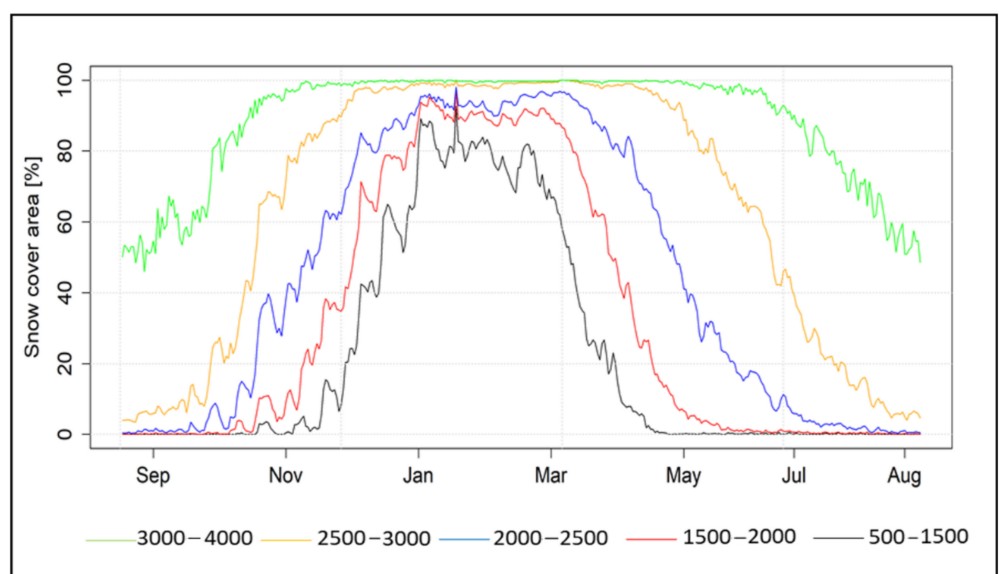

**Figure 4.** Elevation zone-based snow cover evolution with 1000 m intervals.

Figure 5 demonstrates the evolution of characteristic years of snow coverage from 2000 to 2021 and the necessary information. For example, in the hydrological year 2000–2021, the lowest snowfall was in the years 2006–2007, the period of lowest snow cover in the Naryn river basin resulted in extreme water scarcity in 2007, while the highest level of snow cover was in the years 2002–2003.

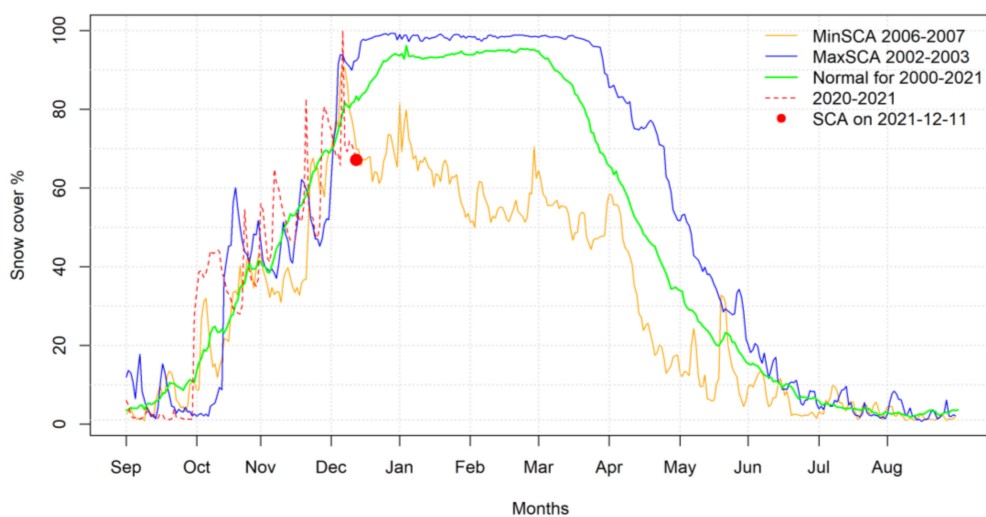

**Figure 5.** Comparative evolution of snow coverage in the Naryn river basin from 2000 to 2021.

It is possible to make a very useful qualitative forecast of river flow for the annual spring and summer seasons considering the snow cover evolution. Additionally, the dry season can be recognized and anticipated early on based on the extent of snow cover during the winter. Besides the comparative analysis of snow coverage, the MODSNOW-

Tool generated spatial snow coverage and temporal evolution for the current hydrological year, as shown in Figure 6, which illustrates the spatial distribution.

### Snow cover for the Naryn basin (2021-12-11)

Total snow cover: 67.1%

| Elevation zone [m] | Snow cover [%] |
|---|---|
| – 1000 | 22.81 |
| 1001 – 1500 | 10.55 |
| 1501 – 2000 | 43.51 |
| 2001 – 2500 | 49.87 |
| 2501 – 3000 | 63.49 |
| 3001 – 3500 | 79.93 |
| 3501 – 4000 | 91.23 |
| 4001 – 4500 | 94.10 |
| 4501 – 5000 | 93.35 |

**Figure 6.** Spatial map of snow coverage in the Naryn river basin on and temporal evolution of snow cover area.

It can be seen that the majority of the Naryn river basin (approximately 65% of the total basin area) is covered by snow. Figure 6 shows the daily report for the Naryn river basin, exemplarily for 11 December 2021. It consists of the snow fraction time series for the hydrological year until the current day (red point in Figure 6); its total snow cover fraction includes its distribution across different elevation zones. Looking at Figure 6 allows making a visual comparison of the water balance; for example, October 2021 had less snowfall than October 2020, but the most impacted month is the last month of snow accumulation (late March or April), which also depends on the high-altitude zone of the study area.

### 3.2. Snow Cover-Runoff Model Set Up

Figure 7 shows the linear regression models for the hydrological forecast for the growing season at three hydrological stations Uch-Terek, Naryn, and Big-Naryn. The linear relationship uses four predictors to forecast models in the growing season (Figure 7). Discharge, precipitation, mean temperature, and mean snow cover for the entire river basin in the last month before the publication of the forecast was used as the main predictors with correlation coefficients (R) of 86, 88, and 91 and coefficient of determination $R^2$ of 0.81, 0.75, and 0.77 (Naryn, Big-Naryn, and Uch-Terek, respectively), showing promising results from forecast models. Since all model simulations rely on the same data sources, a given catchment may have lower model performance. A discrepancy between the physical backbone of the model and the hydrological processes may be prevalent in that catchment [31]. Figure 7 shows a low correlation for 2002 in entire sub-basins (Uch-Terek,

Naryn, and Big-Naryn). A medium correlation was observed in 2019, 2016, and 2007 (Uch-Terek, Big-Naryn), 2016, 2006, and 2008 (Naryn), 2006, 2016, and 2001 (Big Naryn). Projected river discharge diapasons were 123–183 m$^3$ for Naryn, 47–39 m$^3$ for Big-Naryn, and 180–210 m$^3$ for Uch-Terek.

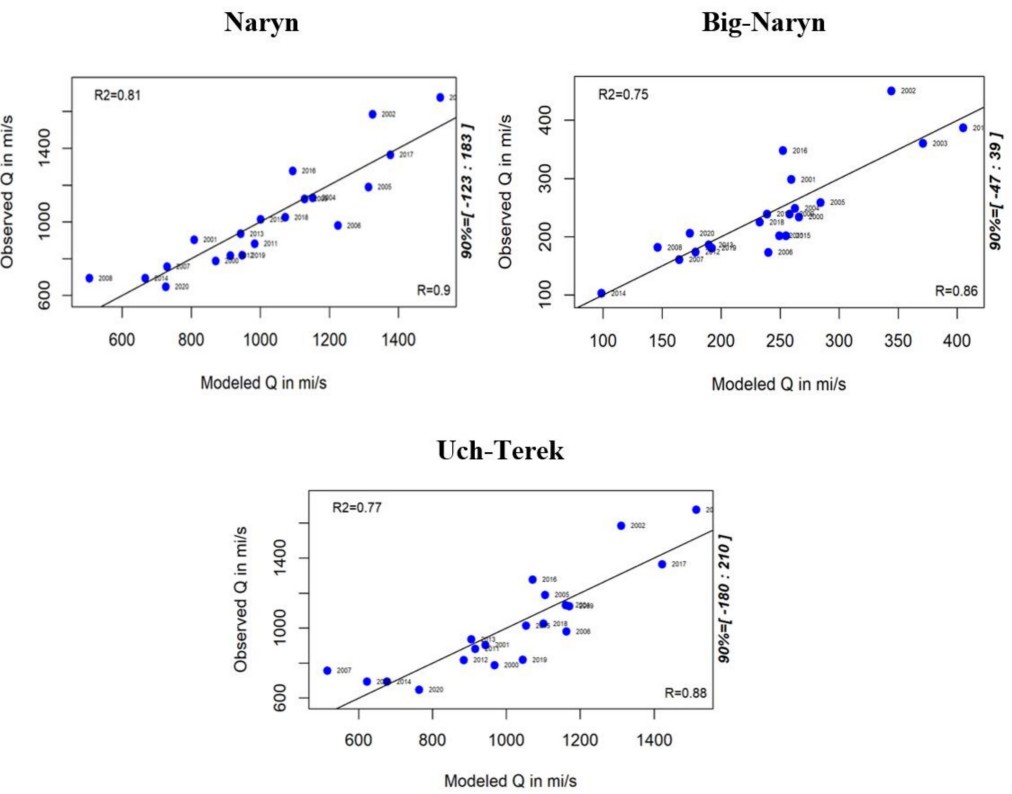

**Figure 7.** Linear forecast models for selected hydrological stations for the growing season.

It can be used to determine the snow cover predictor for the monthly forecast. This graph shows the correlation between each month's water discharge and the snow cover area in each elevation zone. As shown in Figure 8, different correlation values are obtained for different months and altitude zones. River discharge data from three stations with dissimilar physiographic and climatic regions were used, namely station Naryn, from the middle part of the Naryn river basin, station Big-Naryn from the upper basin area, and station Uch-Terek from the lower part of the basin area. Figure 8 showed three stations with different elevation zones, it explained that stations have different altitudes due to geographical conditions that one of the stations has eight elevation zone and the others six zones with 500 m intervals. This is the correlation matrix between snow cover at different elevation zones and monthly discharge. The equations representing these correlations do not give any added value in representing the relationship between these two variables. Therefore, we do add equations as they are already represented by the R values in the matrix figure.

The positive tendencies are mostly present in May and June in the lower part of the basin at station Uch-Terek is 0.60–0.87 at the 2000–3000 m elevation zone. In the middle part of the basin at Naryn station, the correlation shows 0.76 at the 2500–3000 m elevation zone. From May to July, the station Big-Naryn showed the best correlation of 0.70–0.84 at elevation 3000–3500 m in June.

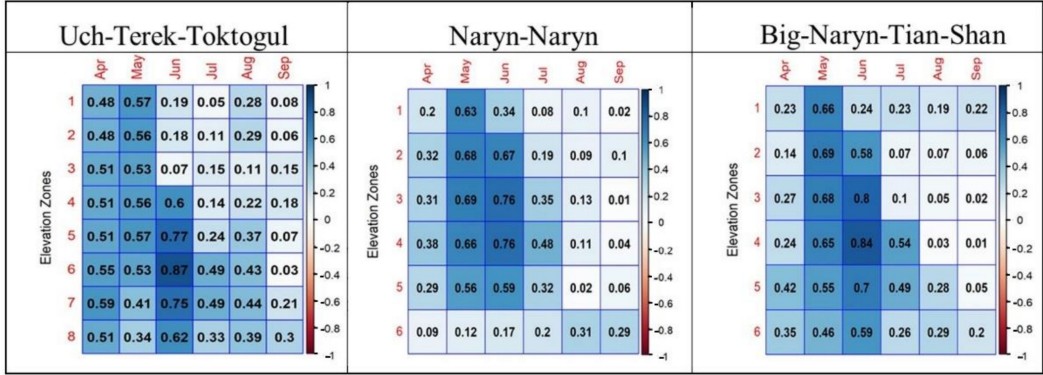

**Figure 8.** Correlation matrix between mean monthly discharge and snow coverage at each elevation zone.

### 3.3. Runoff Distribution Forecast for the Growing Season

The uneven distribution of snow reserves across high-altitude zones in mountain basins and the non-timing of its melting create prerequisites for forecasting of the distribution of runoff by month during the growing season. The distribution of runoff per month during the growing season aims to establish a relationship between the characteristics of snow reserves in the basin and the runoff of each month during growing season. For the monthly forecast during the growing season, data from the previous month is used, and for the growing season (April–September), March data is used as a predictor. This is characterized by the fact that seasonal snow cover from October to March accumulates and determines water runoff for the growing season. Table 1 shows all forecast models in the study area, which is simple but very informative: period (month), the equation for each sub-basin individual. $R^2$, simulated Q, observed Q, the percentage ratio of observed discharge to forecasted discharge, number of confirmed forecasts for the entire period, number of unconfirmed forecasts for the entire period, and the percentage value of confirmed forecasts to unconfirmed forecast.

**Table 1.** Linear forecast models for Monthly and vegetation period (April–September).

| Period | Equation | $R^2$ | SimQ (m³) 2021 | ObsQ (m³) 2021 | Forecast,% | Proved Forecast | Unproved Forecast | Forecast Accuracy 2000–2021 in % |
|---|---|---|---|---|---|---|---|---|
| | **Stations Naryn-Naryn** | | | | | | | |
| Apr. | Apr. = 4.9*SCA − 0.072*Qprev = 0.17*Prec + 1.2*Temp − 434 | 0.70 | 39.3 | 46.2 | 85 | 15 | 5 | 75 |
| May | May = 2.3*SCA + 0.51*Qprev + 0.04*Prec − 0.82*Temp − 78 | 0.60 | 119 | 122 | 98 | 12 | 8 | 60 |
| June | June = 2.9*SCA + 0.88*Qprev − 12*Temp + 70 | 0.62 | 124 | 143 | 98 | 15 | 6 | 71 |
| July | July = 1.7*SCA + 0.29*Qprev + 0.027*Prec + 11*Temp − 95 | 0.54 | 219 | 225 | 97 | 15 | 5 | 75 |
| Aug. | Aug. = −0.66*SCA + 0.37*Qprev − 0.22*Prec − 27*Temp + 674 | 0.25 | 153 | 160 | 96 | 10 | 11 | 47.6 |

**Table 1.** *Cont.*

| Period | Equation | $R^2$ | SimQ ($m^3$) 2021 | ObsQ ($m^3$) 2021 | Forecast,% | Proved Forecast | Unproved Forecast | Forecast Accuracy 2000–2021 in % |
|---|---|---|---|---|---|---|---|---|
| Sep. | Sep. = −13*SCA + 0.54*Qprev + 0.099*Prec − 21 | 0.41 | 78 | 81.6 | 95.5 | 14 | 6 | 70 |
| Apr.–Sep. | Apr._Sep. = 0.018*SCA + 5.5*Qprev + 0.38*Prec + 1.2*Temp − 40 | 0.47 | 129 | 130 | 100 | 13 | 8 | 61 |
| **Stations Big-Naryn-Tian-Shan** | | | | | | | | |
| Apr. | Q_apr = 0.14*SCA_zone_2 + 0.65*Qprev − 0.15*Prec + 0.81*Temp + 30 | 0.34 | 33 | 24.1 | 137 | 13 | 8 | 33 |
| May | Q_apr = 0.31*SCA_zone_2 + 0.61*Qprev − 0.15*Prec + 0.99*Temp + 15 | 0.39 | 50.1 | 62.9 | 80 | 14 | 7 | 33 |
| June | Q_may = 0.15*SCA_zone_2 + 0.87*Qprev + 0.29*Prec − 6.2*Temp − 22 | 0.69 | 68. | 64.3 | 106 | 14 | 7 | 70 |
| July | Q_may = 0.29*SCA_zone_2 + 0.78*Qprev + 0.26*Prec − 5.7*Temp − 27 | 0.71 | 107 | 111 | 96 | 11 | 10 | 55 |
| Aug. | Q_jun = 1.9*SCA_zone_3 + 0.54*Qprev + 0.13*Prec + 4.4 | 0.64 | 71.6 | 72.4 | 96 | 6 | 15 | 65 |
| Sept. | Q_jun = 1.5*SCA_zone_2 + 0.67*Qprev + 0.1*Prec − 4.7*Temp − 15 | 0.67 | 34.5 | 38.2 | 90 | 7 | 14 | 60 |
| Apr.–Sep. | Q_jul = 0.63*SCA_zone_4 + 0.28*Qprev − 0.005*Prec + 61 | 0.43 | 70.4 | 61 | 87 | 12 | 9 | 52 |
| **Stations Uch-Terek-Toktogul** | | | | | | | | |
| Apr. | Apr. = 21*SCA + 1.4*Qprev − 0.14*Prec − 1988 | 0.51 | 296 | 241 | 123 | 15 | 6 | 71 |
| May | May = 9.4*SCA + 1*Qprev − 1.9*Prec − 5.6*Temp + 1 | 0.65 | 665 | 650 | 102 | 12 | 8 | 60 |
| June | June = 9.1*SCA + 0.44*Qprev + 1.1*Prec − 6.3*Temp − 305 | 0.68 | 607 | 580 | 105 | 14 | 6 | 70 |
| July | July = 1.7*SCA + 0.29*Qprev + 0.027*Prec + 11*Temp − 95 | 0.42 | 629 | 625 | 96 | 12 | 8 | 60 |
| Aug. | Aug. = −0.66*SCA + 0.37*Qprev − 0.22*Prec − 27*Temp + 674 | 0.62 | 461 | 415 | 111 | 12 | 8 | 60 |
| Sept. | Sep. = −13*SCA + 0.54*Qprev + 0.099*Prec − 21 | 0.65 | 298 | 310 | 96 | 12 | 8 | 60 |
| Apr.–Sep. | Apr._Sep. = 3.9*SCA + 1.5*Qprev + 0.73*Prec + 16*Temp − 272 | 0.53 | 507 | 473 | 93 | 15 | 6 | 71 |

The forecast for the growing season 2021 shows 87% in Big-Naryn station, 93% in Uch-Terek station, and 100% in Naryn station. The forecast for April shows 61–75% and $R^2$ 0.70, 0.38, and 0.51 (Naryn, Big-Naryn, and Uch-Terek, respectively). The flow of the first month of the snowmelt season on Mountain Rivers depends mainly on the water reserves in the snow cover and air temperature, and the main influencing factor is the location of the watersheds, which are located in relatively low mountains, where almost all the snow has time to melt in this month. In the Naryn basin, intensive snowmelt occurs in May. The upper zones of the pools are sometimes breathtaking. In the lower zones, the snow practically disappears in May. Therefore, the runoff forecast for May for the Naryn basin is based only on the water reserves in the snow cover and its characteristics in table $R^2$ show 0.60, 0.60, and 0.65 (Naryn, Big-Naryn, and Uch-Terek, respectively) in all sub-basins 80–102% in percentage, forecast for May 60–65% came true. In June, forecasts for June in the Naryn Basin are usually sufficient for the central snow reserves to melt, but not completely.

Therefore, the main arguments in the prognostic dependencies are the indicators of the remaining water reserves in the snow cover at the end of May. During this period, the water content of the previous period begins to play a significant role not only as an indicator of the melted amount of snow but also as an indicator of the probable intensity of flood recession for those rivers where the flood peak occurs in May or the first ten days of June. The forecast for June shows 67–71% of ability, and the coefficient of determination shows $R^2 = 0.62; 0.69; 0.68$. Discharge percentage: 105, 106, and 98 m$^3$ (Naryn, Big-Naryn, and Uch-Terek, respectively). In July, the most significant water discharges during the growing season were observed in all watersheds. The runoff during this period is determined mainly by the snow reserves remaining on the basins' surface, the basins' uppermost zones.

The forecast for August and September highlights the features of the forecast methodology. On the rivers of the Naryn basin, this period of decline in water flow, the melting of the seasonal snow supply by this time is almost over, and the monthly flow depends mainly on the intensity of the decline and glacier melt. In the basin area above 3700 m, the snow is still melting, and the glaciers are melting immediately. At this time, the share of glacial feeding of high mountain rivers increases. For August, forecasts show 71–461 m$^3$ and $R^2$ 0.25–0.62. The September runoff is due to the depletion of the seasonal water supply; forecasts give a minimum runoff of 34, 78, and 298 m$^3$ (Naryn, Big-Naryn, and Uch-Terek, respectively). The generic procedure presented for evaluating the goodness-of-fit of mathematical models uses a more general formulation of the coefficient of efficiency (Equation (2)), thus allowing the user to compute modified versions of this indicator instead [32].

## 4. Discussion

For seasonal forecasting purposes, the earth observations of snow cover have been emphasized by Dixon and Wilby [33]. It has been proven that the overall accuracy of the cloudless snow cover data set exceeds 90% using indirect and direct verification methods [26]. Another study was used accuracy assessment for the daily snow depth between the satellite data of MODIS and climate stations and additional snow course measurements for the period 2000–2013 [34]. The results show that it allows accurate estimation of snowline elevation without cloud cover. The cloud removal method in remote sensing images over the earth's surface is that each step of the cloud removal algorithm gives the accuracy of using a satellite image [35]. For this purpose, we used the adapted MODSNOW software package for Naryn river basin. The study of water accumulation in the snowpack yielded many results. Twenty-one-year data were processed, and monthly and growing season forecasts were produced. MODIS snow cover had daily datasets of the study area and hydro-meteorological daily data, making forecasts more explicit. Our results revealed a significant relationship linear forecast model for the vegetation period (Figure 7). This relationship was also confirmed by agreeing well with previous studies [36]. In 2006–2007 there was less snow cover; this year, the water in the Toktogul water reservoir (located downstream of the Naryn catchment) was less than in the other years (Figure 3), which was also confirmed by [37]. The high-altitude zones are given an uneven distribution of

snow cover, which is an essential aspect of snow melting. The low zones begin to melt earlier than the high zones. The main snow accumulation is where the high zones are, regulating the entire water balance for the year [38]. Duethmann et al. [30] made separate analyses for each elevation zone, where we used this method and found good results for the monthly hydrological forecast (see examples in Figure 8). Snowline altitude (SLA) is the most sensitive indicator for monitoring climatic behavior among all the cryosphere elements. In this study, the most sensitive zone is 2500–3500 m elevation, because in these areas there was very good correlation in all stations' snowlines [39]. Most studies indicate the hydrological forecasts, especially on the development of sustainable water management in Central Asia, and it is related to agricultural and livestock production, ecosystems, and socio-economic development [40]. It is challenging to consider all impacting factors in the high mountain areas and make a hydrological forecast due to geographical conditions. The study [41] says that in the winter season, water accumulates in solid form, and in spring, with the increase of temperature, it begins to transform into liquid, and this time the water level rises in rivers, which may cause a different hydrological hazard. Several methods have been proposed to improve hydrological forecasts, especially the water accumulation in snow cover. The Naryn river basin is seasonally snow-dominated; more than 65% of total discharge comes from the snowmelt. The peak of model performance is seasonal vulnerability due to climate change effects of the Naryn river basin where short-time hydrological forecasts are required, for example, monthly or for 20 days and 10 days. The method used in this work may enable short-time forecasts, with high-quality meteorological data and snow cover area data.

## 5. Conclusions

This study processed time series of daily snow cover MODIS data for the Naryn river basin between 2000 and 2021 to calculate the snow cover area, daily river discharge, and meteorological data. To analyze these predictors, the MODSNOW tool was applied. Three hydro-meteorological stations' data were used in the study. We observed positive results from the time series for the whole hydrological catchment. The main objective of this study was to produce a hydrological forecast model. To achieve this, hydrological forecast models were produced for the growing season and monthly scale. Based on this data set of the main predictors, various monthly, multi-monthly, and composite predictors for different forecast dates were obtained using multiple linear regression models and correlations with elevation zones.

The highest correlation was from May to July, and the correlation coefficient was in diapasons 0.56–0.87 in the upper and middle streams 3000–3500 m elevation zones and downstream 2000–3000 m elevation zone. When correlating with altitude zones, it should also be noted that the snowmelt in different altitude zones led to different results due to orography and orientation. The good performance of the study was interpreted in terms of which month snowmelt affects the river flow and in which altitude zone. Another performance parameter is the linear regression showing the goodness of fit ($R^2$ = 0.75–0.81), and the range of river flow can be shown from these results in Figure 7. The diapasons of the hydrological forecast of river discharge: Naryn = 123–183 $m^3$; Big-Naryn = 39–47 $m^3$; Uch-Terek = 180–210 $m^3$. The developed models can mitigate hydrological hazards and help water resources management and the Ministry of Emergency Situations.

**Author Contributions:** Conceptualization, M.P.k. and X.C.; methodology, M.P.k. and A.G. (Abror Gafurov); validation, M.P.k. and E.D.; formal analysis, E.O.; investigation, M.P.k. and E.O.; resources, M.P.k., E.O. and X.C.; data curation, M.P.k., E.O. and E.D.; writing—original draft preparation, M.P.k., T.L. and A.G. (Abror Gafurov); writing—review and editing, M.P.k., A.G. (Abror Gafurov), A.G. (Akmal Gafurov), and E.D.; visualization, M.P.k.; supervision, X.C.; project administration, M.P.k., X.C. and T.L.; funding acquisition, X.C. and T.L. All authors have read and agreed to the published version of the manuscript.

**Funding:** This research was funded by the Strategic Priority Research Program of the Chinese Academy of Sciences, Pan-Third Pole Environment Study for a Green Silk Road (Grant No. XDA20060303), the International Cooperation Project of the National Natural Science Foundation of China (Grant No. 41761144079), the K.C. Wong Education Foundation (Grant No. GJTD-2020-14), and Regional Collaborative Innovation Project of Xinjiang Uygur Autonomous Regions (Grant No. 2020E01010).

**Institutional Review Board Statement:** Not applicable.

**Informed Consent Statement:** Not applicable.

**Data Availability Statement:** Not applicable.

**Acknowledgments:** This study is part of the PhD research of Merim Pamirbek kyzy and has been sponsored by the CAS-TWAS (Chinese Academy and the World Academy of Sciences) President's Fellowship for international PhD students awarded to her.

**Conflicts of Interest:** The authors declare no conflict of interest.

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
