# Peer review of "Hydrological Forecasting under Climate Variability Using Modeling and Earth Observations in the Naryn River Basin, Kyrgyzstan"

_water, doi:10.3390/w14172733_

Round 1
Reviewer 1 Report (Previous Reviewer 2)
General comments: The impact of climate change requires improving the quality of hydrological forecasts in the Naryn river basin. This is especially true for the growing season due to the unpredictable climate behavior. This study simulate hydrological forecasts for three different hydrological stations (Uch-Terek, Naryn, and Big-Naryn) located in the Naryn river basin, the main water formation area of the Syrdarya River. Overall, this is a very interesting research topic. However, several issues need to be resolved before a publication to Water.
Major Comments:
1. The biggest drawback of manuscripts is that only the linear regression models were produced for monthly and yearly hydrological forecasts. Other statistical prediction models, such as machine learning models (random forests) are certainly better than linear regression models. It is suggested to add machine learning method for the hydrological forecasting.
2. In the section 1. Introduction, why there is no introduction of hydrological forecasting methods? This is important and should be included.
3. Language or text writing needs to be checked and revised. For example:
L55— “spatially distributed snow cover space significantly improve hydrological models of runoff forecasting”. Delete the “space”
L61-63, “Traditionally, data from snow surveys and snow observations at weather stations in foothills and mountainous areas are used to predict the flow of mountain rivers in Central Asia.” Some references should be included here.
L86, “…Another study is being carried out to show the spatial and temporal variations in snow cover for the period 2000-2015” Here, references are missing. If I understand you correctly, the citation here should be the “Spatiotemporal variation of snow cover in Tianshan Mountains, Central Asia, based on cloud-free MODIS fractional snow cover product, 2001–2015 [J]. Remote Sensing, 2017, 9(10): 1045.”
Similar problems are no longer enumerated one by one.
4. L160-163 For the linear regression equation, it is suggested that the author should consider comparing the sensitivity of snow line altitude (SLA) and SCA, perhaps using the variable snow line height is more suitable to forecast river discharge. The extraction method of SLA can be referred to the follow papers. If it is difficult to implement SLA extraction in a short time, it is recommended to mention it in the discussion section
Krajcı, P., Holko, L. et al., 2014. Estimation of regional snowline elevation (RSLE) from MODIS images for seasonally snow covered mountain basins. Hydrol 519, 1769–1778.
Deng G, et al. Spatiotemporal dynamics of snowline altitude and their responses to climate change in the Tienshan Mountains, Central Asia, During 2001–2019 [J]. Sustainability, 2021, 13(7): 3992.
Tang, Z., Wang, J., Li, H., et al., 2014. Extraction and assessment of snowline altitude over the Tibetan plateau using MODIS fractional snow cover data (2001 to 2013). Appl. Remote Sens. 8 (1), 084689.
Author Response
Dear Reviewer,
We would like to thank you for your deep reading and your useful comments, which helped us to strongly improve the manuscript.
Please find attached the point-by-point responses word file.
Sincerely,
Authors

Reviewer 2 Report (Previous Reviewer 1)
The authors have investigated correlation between mean monthly discharge and environmental data (snow cover, precipitation and temperature) in three high-altitude catchments.
Authors have presented hydrological forecast results that (potentially) show somewhat high correlation between snow cover and discharge. Unfortunately, manuscript lacks proper methodology description and presentation of the results that would support the proposed approach. Therefore, in my opinion, the aim of the paper in current form is not achieved. I suggest that authors completely revise, restructure and rewrite their paper to reflect the effort they have undoubtably put into their research.
Since this paper is resubmitted version of the manuscript submitted earlier this year in the MDPI’s journal Remote Sensing, without notable changes introduced, there is no point in repeating the remarks here. Authors already have them.
Author Response
Dear Reviewer,
We would like to thank you for your deep reading and your useful comments, which helped us to strongly improve the manuscript.
Please, find attached the point-by-point responses word file.
Sincerely,
Authors

Round 2
Reviewer 1 Report (Previous Reviewer 2)
The author has made a good revision and reply
Author Response
Dear reviewer,
We would like to thank you for your careful and constructive reviews.
Sincerely,
Authors
Reviewer 2 Report (Previous Reviewer 1)
Dear authors,
You keep updating your manuscript in small increments, barely addressing any crucial issues that have been pointed out in the review process.
Nevertheless, I believe that paper can be published once the “Methodology” section is updated to reflect the actual analyses conducted by the authors. In the current version, the methodology only vaguely describes multiple linear regression. Methodology does not clearly describe what data for antecedent discharge, precipitation and temperature were used and by what reasoning. It seems that authors have used simultaneous snow cover and discharge data, which of course resulted in large R2 values since there is high discharge and still present snow cover. There is no background information about processes taken into account when selecting data pairs for correlation.
Please elaborate into detail the methodology behind your results that provides insight to the readers for the interpretation of the results.
Author Response
Dear reviewer,
The authors express their deep gratitude to the reviewer for his thorough review of the manuscript. Taking into account all the recommendations and comments of the reviewer, this manuscript was updated and finalized.
The Methodology section has been updated to reflect the actual analysis done by the authors.
Please see the revised version of the manuscript for a detailed and improved methodology section that will help readers understand the interpretation of the results.
Sincerely,
The authors
This manuscript is a resubmission of an earlier submission. The following is a list of the peer review reports and author responses from that submission.
Round 1
Reviewer 1 Report
The authors have investigated correlation between mean monthly discharge and environmental data (snow cover, precipitation and temperature) in three high-altitude catchments.
Authors have presented hydrological forecast results that (potentially) show somewhat high correlation between snow cover and discharge. Unfortunately, manuscript lacks proper methodology description and presentation of the results that would support the proposed approach. Therefore, in my opinion, the aim of the paper in current form is not achieved. I suggest that authors completely revise, restructure and rewrite their paper to reflect the effort they have undoubtably put into their research.
Please find below few remarks for which I believe should help to improve your manuscript before publishing.
General remarks:
State of the art review is too generic – authors use multiple references for areas very loosely connected to their research (e.g. [5,6]), use obscure references in places where more relevant research could be used (e.g. [11]), case specific references (e.g. [12]) or use inappropriate references such as Journal of Medicine (?) [10].
On the other hand, statements relevant for their research, such as “Traditionally, data from snow surveys and snow observations at weather stations in foothills and mountainous areas are used to predict the flow of mountain rivers in Central Asia” are not referenced.
Similarly, important works recognized by the authors, such as [18], “Observations from the space of spatially distributed snow cover significantly improve hydrological models for runoff forecasting” are not adequately introduced. Overall, there is lack of relevant work addressed that would provide background and relevance of the presented research in the context of the research field.
Please expand on the state of the art review and introduce main challenges of the field that you research is fulfilling to provide context for your research.
Aim of the paper, “The study aims to answer the question of how to make a hydrological forecast for the monthly river flow of the Naryn River Basin” is vague and exclusive. Please formulate your aim in the context of broader outreach and more specific contribution. Forecast of monthly flow can be done using numerous approaches, any of them having advantages. What is the advantage of the approach you are suggesting?
Materials and methods section is incomplete:
· Catchment site lacks basic information – location of gauging stations, characteristic flow data, snowfall, rainfall, temperature overview, etc.
· Figure 2, water evaluation – what does that mean, what is the location used for this, and what relevant info does it provide?
· Why was the elevation divided into 9 zones when only one station spans through all of them and other 2 span over 6 zones?
· Table 1 data has no real value – it can be summed up in a single sentence.
·
Methodology does not clearly describe what data for antecedent discharge, precipitation and temperature were used and by what reasoning. It seems that authors have used simultaneous snow cover and discharge data, which of course resulted in large R2 values since there is high discharge and still present snow cover. There is no background information about processes taken into account when selecting data pairs for correlation.
Section results is basically just data readout, lacking reflection on the actual value of the data.
Figure 3 is incoherent – no relevant yearly data is distinguishable from the given plot.
Figure 4 is now divided into 1000m zones, opposed to 500m ones used previously.
Figure 6 shows temporal (?) evolution of snow coverage, but it is only one moment in time? Which one?
For Figure 8 there is no label relating matrix to the corresponding station.
Correlation matrix lacks background information – please present the equation used.
Discussion is filled with introductory data, completely missing constructive discussion on the results and their contribution.
“The low zones begin to melt earlier than high zones” – where can the supporting data be found in the manuscript?
There is a lot of redundant and repetitive phrasing, e.g.
· All previous studies showed that remote sensing data is useful for water resources management (references?)
· The snow covers in the high mountainous countries act as a water reservoir during the winter season (How can it be a reservoir if it is not controlled? Very general and redundant statement.)
· During the summer months, the increase in discharge (Q) was triggered by snow melting (very general and also questionable since snowmelt is probably initiate during spring months)
· Almost in every paragraph authors reflect on the length of dataset used
Conclusions are simple reflection on the R2 values, instead of general applicability of the approach.
Based on the all of the above, the paper is simple case study that currently presents no real scientific value. Please present your results in a way that your approach can be beneficial to users outside of your catchment.
Specific remarks:
Keyword Central Asia should be replaced by Naryn river basin
Last paragraph of the introduction is not referenced according to the template.
Reference [20] is incomplete
Legend on Figure 6
Table 2 heading is incomplete
Reviewer 2 Report
General comments:
The impact of climate change requires improving the quality of hydrological forecasts in the Naryn river basin. This is especially true for the growing season due to the unpredictable climate behavior. This study simulate hydrological forecasts for three different hydrological stations (Uch-Terek, Naryn, and Big-Naryn) located in the Naryn river basin, the main water formation area of the Syrdarya River. Overall, this is a very interesting research topic. However, several issues need to be resolved before a publication to Remote Sensing.
Major Comments:
1. The biggest drawback of manuscripts is that only the linear regression models were produced for monthly and yearly hydrological forecasts. Other statistical prediction models, such as machine learning models (random forests) are certainly better than linear regression models. It is suggested to add machine learning method for the hydrological forecasting.
2. In the section 1. Introduction, why there is no introduction of hydrological forecasting methods? This is important and should be included.
3. Language or text writing needs to be checked and revised. For example:
L55— “spatially distributed snow cover space significantly improve hydrological models of runoff forecasting”. Delete the “space”
L61-63, “Traditionally, data from snow surveys and snow observations at weather stations in foothills and mountainous areas are used to predict the flow of mountain rivers in Central Asia.” Some references should be included here.
L82, L85, References are cited in inconsistent formats
L83, “…mass balance in the glaciers of Central Asia. Another study is being carried out to show the spatial and temporal variations in snow cover for the period 2000-2015” Here, again, references are missing. If I understand you correctly, the citation here should be the “Spatiotemporal variation of snow cover in Tianshan Mountains, Central Asia, based on cloud-free MODIS fractional snow cover product, 2001–2015 [J]. Remote Sensing, 2017, 9(10): 1045.”
Similar problems are no longer enumerated one by one.
4. L59-60 For the linear regression equation, it is suggested that the author should consider comparing the sensitivity of snow line altitude (SLA) and SCA, perhaps using the variable snow line height is more suitable to forecast river discharge. The extraction method of SLA can be referred to the follow papers. If it is difficult to implement SLA extraction in a short time, it is recommended to mention it in the discussion section
Krajcı, P., Holko, L. et al., 2014. Estimation of regional snowline elevation (RSLE) from MODIS images for seasonally snow covered mountain basins. Hydrol 519, 1769–1778.
Deng G, et al. Spatiotemporal dynamics of snowline altitude and their responses to climate change in the Tienshan Mountains, Central Asia, During 2001–2019 [J]. Sustainability, 2021, 13(7): 3992.
Verbyla, D., Hegel, T., Nolin, A., et al., 2017. Remote sensing of 2000–2016 alpine spring snowline elevation in Dall sheep mountain ranges of Alaska and western Canada. Remote Sens. 9 (11), 1157.
Tang, Z., Wang, J., Li, H., et al., 2014. Extraction and assessment of snowline altitude over the Tibetan plateau using MODIS fractional snow cover data (2001 to 2013). Appl. Remote Sens. 8 (1), 084689.